# Effects of Heavy Ion Irradiation on the Thermoelectric Properties of In_2_(Te_1−x_Se_x_)_3_ Thin Films

**DOI:** 10.3390/nano12213782

**Published:** 2022-10-27

**Authors:** Mannu Pandian, Alageshwaramoorthy Krishnaprasanth, Matheswaran Palanisamy, Gokul Bangaru, Ramcharan Meena, Chung-Li Dong, Asokan Kandasami

**Affiliations:** 1Department of Physics, Tamkang University, New Taipei City 25137, Taiwan; 2Department of Physics, Kongunadu Arts and Science College, Coimbatore 641029, India; 3Materials Science Division, Inter-University Accelerator Centre, New Delhi 110067, India; 4Department of Physics & Centre for Interdisciplinary Research, University of Petroleum and Energy Studies (UPES) Dehradun, Uttarakhand 248007, India

**Keywords:** thermoelectric, defects, power factor, SHI irradiation, electrical resistivity

## Abstract

Ion irradiation is an exceptionally effective approach to induce controlled surface modification/defects in semiconducting thin films. In this investigation, ion-irradiated Se–Te-based compounds exhibit electrical transport properties that greatly favor the transformation of waste heat into electricity. Enhancements of both the Seebeck coefficient (S) and the power factor (PF) of In_2_(Te_0.98_Se_0.02_)_3_ films under 120 MeV Ni^9+^ ion irradiation were examined. The maximum S value of the pristine film was about ~221 µVK^−1^. A significantly higher S value of about ~427 µVK^−1^ was obtained following irradiation at 1 × 10^13^ ions/cm^2^. The observed S values suggest the n-type conductivity of these films, in agreement with Hall measurements. Additionally, Ni ion irradiation increased the PF from ~1.23 to 4.91 µW/K^2^m, demonstrating that the irradiated films outperformed the pristine samples. This enhancement in the TE performance of the In_2_(Te_0.98_Se_0.02_)_3_ system is elucidated by irradiation-induced effects that are revealed by structural and morphological studies.

## 1. Introduction

Energy production and the demand for conventional energy resources are major issues in the present generation [1]. Alternative energy resources are needed for economic and environmental reasons. Thermoelectric (TE) materials convert waste heat that is otherwise lost from domestic devices, automobiles, and industrial equipment into useable electricity. Since the electrical properties of metal chalcogenide-based materials can be tuned with ease, they can be used to improve the performance of energy conversion devices. The energy conversion from thermal to electrical energy in metal chalcogenide-based materials has an intensive role in tailoring the thermoelectric performance of the devices. It is well known that the efficiency of thermoelectric materials is evaluated by a dimensionless figure of merit, ZT = $\frac{S^{2}\sigma T}{k}$ [2,3] (where S, σ, T, and k are Seebeck coefficient, electrical conductivity, the absolute temperature at which figure of merit is measured, and thermal conductivity, respectively). Thermoelectric properties can also be determined by an expression called the power factor (PF) as PF = S^2^σ. An increase in the S value leads to a decrease in σ, and a decrease in σ produces a reduction in the electronic contribution to k [3]. The physical factors (S, σ, and T) are interrelated, and improving one leads to effects of the others’ properties. Better thermoelectric materials have larger values of thermopower and low values of thermal conductivity [4,5]. Thermal conductivity is determined by both the transfer of heat by phonons and charge carriers. The improvement of ZT values has been one of the major challenges in the field in recent years and has been tackled largely by controlling the dimensions of materials on the nanoscale. A nanostructured material has a lower thermal conductivity than the corresponding bulk material because it exhibits more phonon scattering and deformation of its density of states, which results in its thermoelectric power [6]. Metal chalcogenide materials, such as SnSe, PbTe, CoSb, and BiTe, have been more effectively used for thermoelectric applications than other equivalent oxide materials because the electronegativity difference between these pairs of elements is low. Additionally, the large atomic weights of the elements in these materials favor low thermal conductivity [7]. Such chalcogenide materials have been prepared through physical methods such as mechanical alloying, thermal evaporation, and sputtering deposition [8,9,10,11].

In_2_(Te_1−x_Se_x_)_3_ is a layered compound belonging to A^III^_2_B^II^_3_ semiconductors, having a disordered structure with cation vacancies, and is a promising candidate for near room temperature (300–400 K) thermoelectric devices. In–Se–Te systems possesses different structural modifications such as α and β, and one-third or two third of indium sub-lattices remain vacant due to the valence mismatches between the anion and cation. Therefore, In_2_Se_3_ and In_2_Te_3_ find good scope due to the availability of various stoichiometric phases from the In–Se and In–Te phase diagram. Its unique properties such as electrical conductivity, mass density, and thermal conductivity could be utilized to design a new thermoelectric device with a combination of metal chalcogenides. Among these chalcogenides, InSe semiconducting material has a tetrahedral bonding structure with one-third of the cation remaining vacant in the material, which complies with the octet rule for sp^3^ hybridization [12]. On the other hand, In_2_Te_3_ has a zinc blend structure with a defect density in the order of 10^21^ cm^−1^, and one-third of the cation sites are vacant due to the vacant sub-lattices between the anion and cation [13]. Further, these vacancies in crystal sites lead to the exceptional reduction in the lattice vibrations owing to strong phonon scattering. Reportedly, the conductivity of In_2_Te_3_ is higher than that of In_2_Se_3_ by considering the bond energy difference between the In–Se and In–Te elements [14]. Though the electrical conductivity of In_2_Se_3_ thin films is about 10^−8^ (Ω cm)^−1^ at room temperature, the thermal conductivity of In_2_Te_3_ crystal is about 1 W m^−1^ K^−1^ at 450 K. Consequently, the deposition of nanostructured thin film could be used to tailor the above values by reducing the dimensionality of the material to the nanoscale regime because of the possible enhancement in the phonon scattering at the grain boundary regions. Most importantly, this metal chalcogenide-based material has large radiation stability of electronic parameters after ionization fluence, and it has been explored up to 10^18^ fast neutrons per cm^2^ [15]. Thus, these materials are much needed for the thermoelectric generators in the zone of nuclear reactors. The efficiency of thermoelectric materials mostly depends on their surface morphology, material defects, stoichiometric composition, and other factors. The ion beam processing of such materials can be used to explore the material–ion beam interactions at the surface/interface level. High-energy heavy ions with velocities comparable to or higher than the velocity of the orbital electron are referred to as Swift Heavy Ions (SHI). SHI irradiation (with an energy of over tens of MeV) is one of the unique approaches to modifying the thin films of thermoelectric materials and also to understanding the ion interaction of solid surfaces and interfaces [16]. For an ion beam to be considered "swift", the ions must have the energy of tens of mega electron volts (MeV) [17]. In the case of thin films, SHI irradiation may induce surface modifications with nanostructuring at the surface of the materials.

At a higher energy regime, the inelastic collision between the projected ions with the target lattice dominates, which causes excitation and ionization of the lattice point [18]. Significant atomic rearrangements/displacements were observed in materials when heavy ions passed through a target material. Hence, high-energy ion beam irradiation on these chalcogenide materials can improve the electrical transport properties to enhance the thermoelectric performance. Depending on the mass and energy of the irradiated ions, a wide range of damage and modifications are generated in the material. In the case of light ions, the displacement cascades will be linear where the atomic motion of the incident ion would immediately stop after a few collisions and thereby creating only isolated point defects such as vacancies and interstitials. The number of locally produced vacancies depends on the ion mass. However, low-energy ions introduce impurities and defects such as oxygen vacancies into a lattice or examine the novel properties resulting from the impurity defect interactions. Thus, oxygen deficiency can influence the thermoelectric properties of nitrogen ion-implanted SrTiO_3_ films [11].

For the same target, heavy ions tend to form denser collision cascades as compared to those produced by light ions and thereby produce a huge number of recoil atoms. Therefore, electronic energy loss is dominant at high-energy ion irradiation, though both low- and high-energy irradiation induces many defects, including point defects, dislocations, interstitials, and vacancies, and thereby strongly influence the electrical and thermoelectric properties [9,10,11]. Heavy ion irradiation on the material results in the formation of point defects or defect clusters that increase with ion fluence; consequently, the mobility of charge carriers across these defects decreases, thereby enhancing thermopower [10]. Here, the creation of point defects and vacancies on irradiation act as carrier traps that could be the reason for the decrease in mobility. Additionally, heavy ion beam irradiation proposes an important means of alloy formation with the creation of defects and nanostructures in the material, which eventually lead to enhanced thermopower and the lowering of the thermal conductivity by carrier energy filtering and phonon scattering, respectively [6,10,19]. SHI passes through the target and predominantly deposits large energy in the electronic sub-system of the material, creating point defects that can be electrically active and enhance the charge transport in the lattice [10]. Heat treatment of the irradiated sample provided ion-generated defects that enhanced compound formation.

Considering the above facts, the ion irradiation of solid-state materials generates nanostructures or defects that enhance thermoelectric power and thus diminish thermal conductivity by phonon scattering. Masarrat et al. investigated the effect of Fe ion implantation on a CoSb_3_ system and reported a high power factor of Fe-implanted samples owing to the creation of vacancies at Co sites [9]. Bala et al. enhanced the thermoelectric properties of CoSb alloy by irradiating it with a 100 MeV Ag ion [10], and they found that ion beam mixing followed by post-annealing samples showed improvement of such properties compared to the thermally annealed samples. Bhogra et al. examined the electrical transport properties of nitrogen ion-implanted SrTiO_3_ films and found that the power factor increased with the ion fluence [11]. Chien et al. reported the enhanced thermoelectric properties of gallium ion-irradiated Bi–Sb–Te nanowire (NW) and identified size-dependent thermoelectric power and ZT values by introducing the phonon-scattering of defects [19]. The NW size was reduced intensely, and the thermal conductivity showed a dramatic 74% drop due to the substantial formation of an amorphous-like structure. Their report further suggests that both the electrical and thermal conductivities were reduced intensely due to the phonon scattering of defects by Ga ion irradiation. The observed difference was attributed to the presence of more vacancies and disorders created during the ion beam irradiation process. Recently, Ma et al. investigated the 320 keV Fe^10+^ irradiation-induced modification of CuAlO_2_ films and their improved TE performance, which was caused mostly by the formation of copper vacancies in irradiated films [20]. Kim et al. studied the influence of an He^+^ ion beam on the MoS_2_ system and found that irradiation-induced defects significantly increased its power factor [5]. SHI irradiation is a novel technique for forming nanostructures/microstructures and has been used to alter the properties of a material for its use in the fabrication of devices. It also produces several interesting phenomena, such as thermal spiking, the rearrangement of interfacial atoms, the formation of vacant lattice sites, grain fragmentation, and grain alignment [9,10,11,21,22,23].

In a previous study, the effect of thermal annealing on Se-doped In_2_Te_3_ thin films and their thermoelectric properties was investigated [24], and thermal annealing resulted in an improvement in the PF value compared to the as-prepared film. SHI irradiation influences the electrical transport properties due to grain boundary scattering, as evidenced by the previous studies discussed above. Ni ions were chosen primarily since they are heavy ions that produce large collision cascades without the need for very high doses of irradiation. Considering these facts, the present work attempts to evidence the effects of heavy ion irradiations and to explain the microstructural changes in terms of the grain fragmentation/recrystallization processes in the material. The effects of grain boundary scattering on the thermoelectric properties of the In_2_(Te_0.98_Se_0.02_)_3_ system could also be one of the leading factors to be investigated. In this work, the effects of 120 MeV Ni ion irradiation on the In_2_(Te_0.98_Se_0.02_)_3_ alloys and their thermoelectric properties were explored.

## 2. Materials and Methods

In_2_(Te_0.98_Se_0.02_)_3_ thin films of 700 nm thickness were deposited on the glass substrate using thermal evaporation at a pressure of 2 × 10^−6^ mbar. The optimized composition was prepared by taking In of 0.6045 g, Te of 0.98737 g, and Se of 0.01262 g. In other words, 1 g of Te would combine with 0.5999 gm of In and 1 g of In would combine with 0.9873 gm of Te. Then, 1 g of (Te_0.98_Se_0.02_)_3_ would combine with 0.6045 gm of In_2_ in this In_2_(Te_1−x_Se_x_)_3_ system. X-ray diffraction (XRD) patterns of the films were obtained to determine their structural parameters. Atomic force microscopy (AFM) and field emission scanning electron microscopy (FESEM) were used to obtain surface micrographs of the samples. The electrical transport properties such as electrical resistivity (*ρ*) and thermoelectric power (S) of the In_2_(Te_0.98_Se_0.02_)_3_ films were obtained in the temperature range of 300–400 K using a standard DC four-probe and the bridge method [25]. Hall effect measurements were performed using an ECOPIA HMS-3000 system in Van der Pauw configuration; this measurement provided the Hall coefficient and other paraments such as carrier concentration (n), Hall coefficient (R_H_), carrier mobility (µ), resistivity (*ρ*), the type of conductivity (n or p), etc. The prepared In_2_(Te_0.98_Se_0.02_)_3_ films were irradiated with 120 MeV Ni^9+^ ions at ion fluences of 1 × 10^11^ ions/cm^2^ to 1 × 10^13^ ions/cm^2^ at a current of 0.50 pnA (particle nano Ampere) using a 15 UD Pelletron at the Inter University Accelerator Center (IUAC), New Delhi, India, and the projected ions were scanned over the targets using an electrostatic scanner. The ion fluences of 1 × 10^11^ ions/cm^2^, 1 × 10^12^ ions/cm^2^, and 1 × 10^13^ ions/cm^2^ samples were named as 1E11, 1E12, and 1E13, respectively.

Figure 1 displays the calculated electronic energy loss (S_e_) using SRIM simulations, and the respective values are presented in Table 1. For the Ni^9+^ ion irradiation, the value of the electronic energy loss (S_e_—due to inelastic collision) and nuclear energy loss (S_n_—due to elastic collision) was calculated using SRIM-08 and was about 1110 eV/Å and 2.06 eV/Å, respectively. The projected range of 120 MeV Ni^9+^ ions in the films was also calculated and was 16.00 µm, which was greater than the film thickness (700 nm). Thus, the bombarding Ni ions passed through the whole sample and were mostly deposited in the glass substrate.

## 3. Results and Discussion

### 3.1. Microstructural Analysis

Figure 2 shows the XRD patterns of pristine and irradiated In_2_(Te_0.98_Se_0.02_)_3_ films. The peak positions at 2θ = 40.69° corresponded to the (21 $\overset{-}{1}$
$\overset{-}{4}$) plane of the In_2_(Te Se)_3_, agreeing with the standard JCPDS diffraction data (File number 76-1182). Comparing the pure In_2_Se_3_ and In_2_Te_3_ phase, a peak at 23.15° assigned to the pure In_2_Te_3_ phase of the film exhibited the preferential orientation of the (5 1 1) plane (JCPDS 16-0445). Further, the peaks at diffraction angles 2θ = 27.68° and 38.53° were assigned to the (0 0 6) and (1 2 0) planes, respectively, of γ-In_2_Se_3_ in pristine and irradiated films (JCPDS file number 71-0250). However, the formation of pure In_2_Se_3_ and the In_2_Te_3_ phase in the present work was compared with the available literature. Matheswaran et al. prepared InTe thin films and reported the existence of mixed phases in XRD patterns that include In_2_Te_3_ and In_2_Te_5_ phases [26,27]. Niranjan et al. prepared InSe thin films and showed the presence of InSe, In_4_Se_3_, and γ-In_2_Se_3_ phases. Comparing these results with the present work, the formation of pure γ-In_2_Se_3_ and In_2_Te_3_ phases along with preferential orientation also agreed with this literature. The reason for the formation of the In_2_Se_3_ and In_2_Te_3_ phase compounds can be simplified by considering the electronegativity difference among the choice of materials. The elements having high electronegativity differences may have high binding strength [28], and the elements with a small difference in electronegativity may induce poor stability of a compound [29]. For instance, the elements In, Te, and Se have 1.78, 2.10, and 2.54 Pauling, respectively. Among these elements, the electronegativity difference is higher for the In–Se system (0.76) compared to those of the In–Te (0.32) and Se–Te (0.44) systems. By considering these facts, we assume that there may be a possible chance to form a dominant In_2_Se_3_ phase primarily due to the high electronegative difference (0.76) and the rest of the compounds later (other possible phases).

The irradiation-induced structural modification in 1E13 slightly increased its peak intensities over those of the pristine, 1E11, and 1E12 samples. The phase formation was almost same for all samples. The only difference was the shift in diffraction angle and variation in the peak intensity. Notably, the lack of a peak that corresponded to the extra crystalline phase or any metallic peak (indium) in the XRD pattern revealed the stability of In_2_(Te_0.98_Se_0.02_)_3_ films under irradiation and also the lack of amorphization in the sample, even at 120 MeV Ni ion irradiation. Hence, the S_e_ of 1110 eV/Å did not induce amorphization in the In_2_(Te_0.98_Se_0.02_)_3_ system.

The essential structural parameters, namely, crystallite size (D) (calculated using Debye–Scherrer [30]), dislocation density (δ) [31], and strain (ε) [32], were estimated by the following equations:D = kλ/βcosθ (nm)(1)
δ = 1/D^2^ Lines/m^2^(2)
ε = β/4 tanθ(3)
where the constant k is the shape factor = 0.94, λ is the wavelength of X-rays (1.5406 nm for CuKα), θ is the Bragg angle, and β is the full width at half maximum of the diffraction peak measured in radians.

The crystallite size was determined, and the average crystallite size was about 23.5 nm for the pristine sample. The initial ion fluence produced a small increase in the crystallite size to 24.60 nm owing to the ion beam-induced effect below the threshold value of S_e_, resulting in the crystallization of the materials. The decrease in the crystallize size may have been due to strain-induced fragmentation of crystallites by the influence of SHI irradiation. A similar grain growth mechanism was also reported by Kumar et al., namely, 120 MeV Au^9+^ ion beam-induced modifications of SnO_2_ and TiO_2_ nanocomposite thin films with varying fluence from 5 × 10^11^ ions cm^−2^ increasing to 2 × 10^13^ ions cm^−2^ [21]. Crystallite sizes were reduced to 19.36 nm and 17.15 nm for ion fluences of 1E12 and 1E13, respectively. This reduction in crystallite revealed the occurrence of grain fragmentation/recrystallization processes in the material. The prominent decrease in the crystallite size with the increase in ion fluence was due to the irradiation-induced lattice defects in the material. Mostly, the crystallite growth/grain fragmentation occurring in polycrystalline samples was due to the spread of energy loss by SHI irradiation. Additionally, this reduction in crystallite size could be attributed to the strain-induced grain fragmentation of crystallites as well. Based on the thermal spike model [22] and the Coulomb explosion model [33], the irradiation-induced defects/strain in the crystallites could be described.

In the case of the thermal spike [22], SHI passed through the target and deposited large energy in the electronic sub-system of the material, creating defects. The highly excited electron quickly distributed its energy through electron–electron interaction until electrons were thermalized to dissipate their energy through electron–phonon coupling to the atoms in the target material. This energy was highly excited by electron–electron coupling and transferred to the lattice atoms. Thus, SHI irradiation through the system or target produced a large increase in lattice temperature that may have led to strain in the crystallite. The produced strain may have caused fragmentation in crystallites. The local temperature during the thermal spike phase increase up to ~10^4^ K spread and quenched subsequently within ~10^−12^ s. The resulting liquid-like non-equilibrium state induced various thermally activated processes, such as atom migration, evaporation of atoms, and atomic jumps across grain boundaries. This resulted in permanent structural and surface modifications that included defects or ion track phase transitions [9,10]. The coulomb explosion model explains that the ionized zone of the positively charged particle is enormously produced over the path of the incident ion by electrostatic repulsive forces, which ultimately induce strain in the system inside. Moreover, the existence of strain in the system by SHI irradiation may assist in fragmentation in the crystallites. The significant increase in dislocation density (δ) suggests notable damage on the surface as well and is consistent with the results obtained from the morphological analysis, as larger grains were fragmented to smaller size grains. The dislocation density in 1E11 was lower than that in the pristine sample but increased with the ion fluence, suggesting more grain fragmentation in material under a higher ion fluence. The widening of the peaks in the XRD patterns was attributable to the decrease in crystallite size, suggesting the presence of lattice defects/strain [21]. This decrease in the crystallite size originated from the defect structure induced by the ion beam in the material. The observed results were similar to those reported by Panda et al., namely, 140 MeV Ni ion-irradiated AgInSe_2_ and Ag_2_Se composite thin films [34].

### 3.2. Morphological Studies

The pristine thin film shown in Figure 3a exhibits the presence of a smooth surface morphology with spherical grains, which is different from that of the 1E11 sample (Figure 3b). The increase in ion fluence to 1E12 led to grain fragmentation, as larger spherical grains became smaller grains. The grains in 1E13 seemed to be smaller and denser, being significantly different to those in the pristine thin film. High-energy deposition on target materials reduced the larger grains into smaller grains, and this behavior was consistent with the structural analysis, as crystallite size decreased with ion fluence. Grains are composed of many crystallites. In the present case, it seems grains observed in SEM images were also composed of many crystallites with a size of about 20 nm. The observed behavior is similar to the result reported by Zara et al. on the influence of 100 MeV O^7+^ and 100 MeV Si^7+^ ion irradiation on indium thin films and explains irradiation-induced grain fragmentation in these materials [23]. They found that heavy ion irradiation results in larger size grains becoming fragmented into smaller grains with an increase in the ion fluence for both O^7+^ and Si^7+^ ion species. Increasing the ion fluence reduces the crystallinity of the film due to the irradiation-induced grain fragmentation.

The elemental ratio for In:Te:Se is 40%:58%:2%. The elemental compositions of the films were analyzed by using the EDS technique. The actual elemental compositions of the films are displayed in Table 2. A slight variation in the elemental composition of irradiated samples was observed compared to that of pristine film, which may have been because of the irradiation-induced high energy deposition in the materials by Ni ion beams. This deviation is clarified as follows: The pure crystalline/amorphousIn_2_Se_3_ phase was detected for all samples. From the calculated and measured elemental compositions, the Se and Te elemental ratio confirmed a smaller deficiency compared to that of In. Further, it also proved that because of the binding energy difference among these elements, the formation of the InSe phase was more comparable to those InTe phases. Moreover, the In ion evaporated significantly during the thermal processes due to the difference in atomic vapor pressure. The calculated and measured elemental ratios between In, Te, and Se are presented in Table 2. Apart from In, Te, and Se peaks, a small amount of Si, C, and O peaks from the glass substrates were observed (i.e., we used the normal and distilled water to rinse the slides as well as agitate for cleaning purposes), confirming the nature of the prepared In_2_(Te_1−x_Se_x_)_3_ thin films.

Figure 4 presents a change in the AFM micrographs of In_2_(Te_0.98_Se_0.02_)_3_ as a result of ion irradiation. The micrograph of the pristine film revealed a smooth surface with spherical-shaped grains. The estimated average (R_a_) surface roughness and root mean square (R_q_) roughness of the pristine sample were about ~12.65 nm and ~16.23 nm, respectively. The observed R_a_ and R_q_ increased to ~13.54 nm and ~17.37 nm, respectively, for 1E11. However, R_a_ decreased to 10.18 nm and R_q_ to14.52 nm for 1E12 and 8.13 nm and 12.24 nm for 1E13. The grains were found to be uniformly distributed throughout the entire surface of the pristine sample, and their average size was ~30 nm. In 1E11, it slightly increased to ~32 nm. However, it decreased to ~27 nm and ~24 nm in 1E12 and 1E13, respectively. Thus, AFM studies proved surface smoothening as the ion fluence increased, possibly because of irradiation-induced viscous flow in the sample, as explained by Zhang et al., on Ti-based bulk metallic glass by heavy ion irradiation [35]. Further, the mean surface roughness became reduced with an increasing 20 MeV Cl^4+^ ion dose, which confirmed the smoothing of surfaces. It is believed that the process of surface smoothing and roughing in solids is rather complex, depending on the properties of the incident ion, incident angle, and the target materials. In the case of low-energy ion irradiation, the projectiles are implanted the near surface. At higher fluence, the key reason for surface roughening is likely surface erosion or deposition, whereas surface smoothing seems to be due to surface diffusion or viscous flow.

Therefore, ion beam-induced fragmentation or recrystallization reduces grain size under high energy irradiation, which causes melting followed by recrystallization. From the AFM image, the observed average grain size was about 30 nm for the pristine sample, i.e., the scanned image size was larger in AFM (5 µm). While AFM gave more information about these surfaces, the SEM image provided details of the grains. Grains were composed of many crystallites, and the estimated grain size in the SEM image was about 100 nm, as pointed out in Figure 4. In the present case, it seems that grains observed in SEM images were also composed of many crystallites with a size of about 20 nm. Further, it is clear that both SEM and AFM showed very similar kinds of spherical shaped images with different magnifications, and also it seemed that larger grains were composed of many crystallites for the pristine sample.

### 3.3. Measurement of Seebeck Coefficient (S) and Electrical Resistivity

The Seebeck measurement is performed by creating a temperature difference using the differential method [25]. The average temperature difference is measured by creating the temperature difference in both directions and taking the average of both. The resistivity is measured in a four-probe mode. Figure 5 presents the measured S of In_2_(Te_0.98_Se_0.02_)_3_ at 300 K to 420 K temperature. The negative S value suggested the dominance of electron charge carriers in all samples. The observed S value for the pristine sample was ~221 µVK^−1^. The S value was found to increases with ion fluence to about ~427 µVK^−1^ for 1E13. This result indicated that S increased with increases in the ion fluence to a maximum (1E13), showing double the value of the pristine sample. Notably, S increased linearly with temperature for In–Se–Te, suggesting possible semiconducting behavior. The improved S value was evidence of an enhanced PF value in irradiated In_2_(Te_0.98_Se_0.02_)_3_ films. The observed S values consistent with the measured S of the n-type of In–Se-based chalcogenides reported by Dhama et al., were about −159 µVK^−1^ to −568 µVK^−1^ [8]. However, not much work has been explored on the thermoelectric performance of In_2_(Te_0.98_Se_0.02_)_3_ alloys, especially under SHI irradiation.

Figure 6 displays the measured PF for pristine and irradiated In_2_(Te_0.98_Se_0.02_)_3_ thin films. The maximum power factor at 400 K was found to be ~2.37 µW/K^2^m for 1E11, which displayed a higher PF value than the pristine sample (~1.23 µW/K^2^m). The PF value for 1E12 was about ~3.72 µW/K^2^m, better than that of the pristine sample. The PF value further increased to ~4.91 µW/K^2^m for 1E13, which was four times higher than that of the pristine sample. The effect of heavy ion irradiation on In_2_(Te_0.98_Se_0.02_)_3_ samples demonstrated considerable enhancement of both S and PF, which is evidence of the presence of more SHI-induced defects in the irradiated samples than in the as-deposited films. The PF values were found to exponentially increase with the ion fluence, which showed considerable enhancement in the irradiated samples as compared to the pristine samples. The higher value of the S led to an increase in the PF values of the irradiated samples. The higher S value for the irradiated sample was possibly because of grain fragmentation from large nanograins into tiny nanograins, as evidenced from FESEM analysis. Further, irradiation created defects inside the material, and these defect centers helped in charge carrier filtering, increasing the Seebeck coefficient (S), while these defects hindered the motion of charge carriers in conductivity measurements, leading to decreases in conductivity with irradiation. The heavy ion irradiation in the material resulted in the formation of point defects or defect clusters, which increased with ion fluence, and consequently the mobility of charge carriers across these defects decreased, thereby causing enhancement in thermopower. Bala et al. also reported a similar result that enhanced the thermoelectric properties of CoSb_3_ alloy by 100 MeV Ag ion irradiation [10].

The existence of a high density of nanoscale grain boundaries might also lead to improvements in thermoelectric properties of In_2_(Te_0.98_Se_0.02_)_3_ thin films. The observed results also agree with earlier reports that suggest that phonon scattering through the nanoscale grain boundary leads to significant thermoelectric enhancement [10,36,37]. Sanyal et al. studied the impact of grain boundary scattering in polycrystalline CuInSe_2_ films, and their results suggest that electrical conductivity, Hall mobility, and carrier concentration are influenced by the dominant grain boundary scattering effects [38]. Consequently, the effect of grain boundary scattering in the material could also lead to an impact on the electrical transport properties of the In_2_(Te_0.98_Se_0.02_)_3_ films. The presence of interface states along with thermionic emissions across the grain boundaries directly affects the charge transport mechanism in polycrystalline films. Wu et al. reported similar studies that prove that the formation of nanoscale grains can reduce the lattice thermal conductivity dramatically in nanocrystalline PbS materials, and further theoretical modelling also confirmed that high densities of nanoscale grain boundaries were more effective in reducing lattice thermal conductivity, thereby improving thermoelectric performance [36]. Additionally, Poudel et al. also reported bismuth antimony telluride-based alloy compounds, which showed excellent thermoelectric properties due to the low thermal conductivity caused by the improved phonon scattering by nanograin boundaries and defects [37]. By considering the above facts, it seems that the effect of grain boundary scattering in the present In_2_(Te_0.98_Se_0.02_)_3_ system could also be one of the leading factors in the enhancement of the thermoelectric properties of the materials.

In general, electrical resistivity (*ρ*) and electrical conductivity are related to carrier concentration (n) through carrier mobility (µ):σ = 1/*ρ* = neµ (4)

Both the carrier concentration and carrier mobility contribute to the conductivity.

Further, the relation between thermoelectric power (S) and carrier concentration can be explained by the following Mott mathematical formula:(5)$$S=\frac{8\pi^{2}k^{2}B}{3{\mathrm{eh}}^{2}}m^{*}T{\left(\frac{\pi }{3n}\right)}^{2/3}$$
where kb is the Boltzmann constant, h is Planck’s constant, *m** is the effective mass of carriers, and *T* is the absolute temperature. The above relation (5) proposes that the S value generally depends on the carrier concentration. Thus, increases in carrier concentration and ion fluence considerably reduce the S value, and S has been revealed to be inversely proportional to 2/3 power of carrier concentration from the above relation. However, an increase in ion fluence shows a significant increase in S value, owing to the SHI-induced defects in the material. The carrier concentration is not only the major reason for the variation in the S value. Addition factors such as defect level and charge scattering in grain boundaries are also defining properties of the materials. Hence, in the present case, ion beam-induced nanostructuring played a significant role in the improvement of S.

Figure 7 plots the measured electrical resistivity (*ρ*) of In_2_(Te_0.98_Se_0.02_)_3_ thin films and confirms the semiconducting behavior of the materials [39]. The *ρ* curve seemed to be higher for irradiated samples than for pristine samples, revealing the influence of SHI irradiation, which modified the surface of the materials through the grain boundaries. This phenomenon may be responsible for the decrease in the carrier mobility of the In_2_(Te_0.98_Se_0.02_)_3_ system. Table 3 presents the Hall measurements of pristine and all irradiated In_2_(Te_0.98_Se_0.02_)_3_ samples. The negative Hall coefficients indicated an n-type conduction of the films, as presented in Figure 8b. Measured carrier concentrations were about 1.21 × 10^17^ cm^−3^ to 3.927 × 10^18^ cm^−3^, as shown in Figure 8a.

Table 3 displays the resistivity, carrier concentration, and Hall mobility of pristine and irradiated samples. The carrier concentration for the pristine sample was found to be 1.21 × 10^18^ cm^−3^. The carrier concentration at different ion fluences of 1E11, 1E12 and 1E13 were found to be about 8.35 × 10^17^, 17.24 × 10^17^, and 39.27 × 10^17^ cm^−3^, respectively (shown in Figure 8). The improvement in carrier concentration with an increase in the ion fluence may have been due to increases in defects such as vacancies or interstitial defects. The carrier concentrations were higher at room temperature for the irradiated films as compared to the pristine sample, which led to an increase in the electrical resistivity of the irradiated samples due to the decrease in the mobility of the charge carriers. Mostly, ion beam irradiation enhanced the resistivity due to the creation of defects. Hence, irradiated samples showed higher resistivity and revealed the presence of higher carrier concentrations and minimum carrier mobility that may have been a result of grain fragmentation. The higher carrier concentration in the irradiated samples also confirmed the presence of more defects.

However, the carrier mobility decreased as the ion fluence increased, and this effect may have been responsible for the higher *ρ* of the irradiated samples, owing to grain fragmentation. Likewise, the *ρ* curve increased, possibly because the ion irradiation created defects/impurities, reducing the mobility of carriers in the materials. Furthermore, the small grains favored electron scattering by grain boundaries, thereby enhancing the *ρ* of the materials. The greater ratios of surface area to volume in small grains may have led to a greater ratio of grain boundary to dislocations being produced. Consequently, the *ρ* increased with irradiation as grain size decreased, suggesting low electrical conductivity. Samples with small grains contain more grain boundaries and scatter electrons, thereby increasing the *ρ* of the material. The observed result is similar to the observed case of n-type (Hf, Zr) CoSb thermoelectric material, demonstrating the grain size-dependent electrical properties that reveal the dominant grain boundary scattering mechanism [40]. Further, some trapped states at the grain boundary may also contribute to the higher resistivity of the materials. The larger grain boundary region has an additional scattering center in In_2_(Te_0.98_Se_0.02_)_3_, which could be a reason for higher *ρ* in irradiated samples than in the pristine sample. The presence of several scattering centers limits the *ρ* of the materials, consistent with FESEM analysis. Then, the number of scattering centers leads to a rapid increase in electrical resistivity. Biswas et al. reported that the thermal conductivity of Al-doped ZnO quantum dots is reduced due to the selective phonon scattering by point defects and interfaces, and thereby an enhanced PF value [41]. Further, the lattice thermal conductivity of this composite could be dominated by grain boundaries and hence phonon–phonon scatterings. Therefore, grain boundary scattering may have a strong influence on defining the Seebeck coefficient of the materials. Accordingly, the enhancement in thermopower for the irradiated In_2_(Te_0.98_Se_0.02_)_3_ films could also be attributed to charge scattering due to the grain boundary.

Additionally, point defects also could be one of the determining factors of impurity scattering results in the enhancement of thermopower. The high thermopower value for the irradiated sample is mostly because of the contrast in charge scattering due to the grain boundary. Ion irradiation creates defects and vacancies, which affect the transport properties of the carriers. The observed increase in S could be accounted for by the modification of these properties under irradiation. Moreover, an increase in ion fluence shows a significant increase in S value that may be due to the SHI-induced defects in the material.

## 4. Conclusions

The thermoelectric performance of ion-irradiated In_2_(Te_0.98_Se_0.02_)_3_ alloy was investigated, along with its structural and surface morphologies. All irradiated and pristine samples exhibited negative S values. The S value of ~427 µVK^−1^ for 1E13 was double the value of the pristine sample (~221 µVK^−1^) at 400 K. The PF values of all irradiated samples were better than that of the pristine sample. FESEM and AFM micrographs of irradiated samples revealed small grains, suggesting grain fragmentation as a result of ion irradiation. Substantial improvements in the thermoelectric performance of the material by high energy ion irradiation was caused by the formation of defects, as evidenced by structural and morphological studies. The electrical transport properties of the In_2_(Te_0.98_Se_0.02_)_3_ compound suggest that it may be an efficient material for use in thin film-based thermoelectric devices.

## Figures and Tables

**Figure 1 nanomaterials-12-03782-f001:**
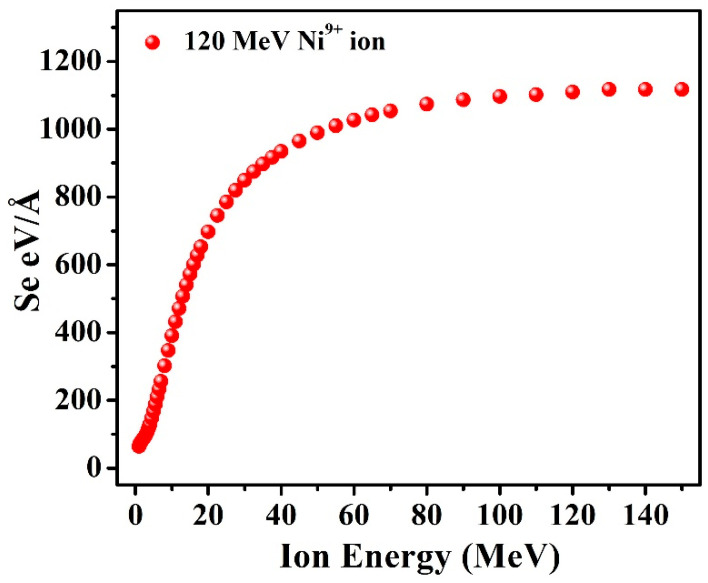
The calculated electronic energy loss (S_e_) using SRIM simulations.

**Figure 2 nanomaterials-12-03782-f002:**
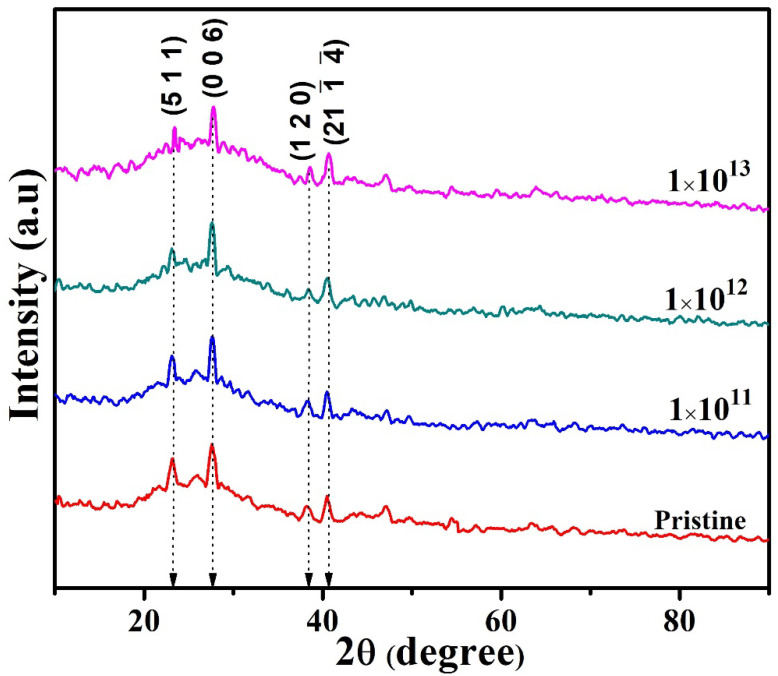
XRD patterns of pristine and irradiated In_2_(Te_0.98_Se_0.02_)_3_ films.

**Figure 3 nanomaterials-12-03782-f003:**
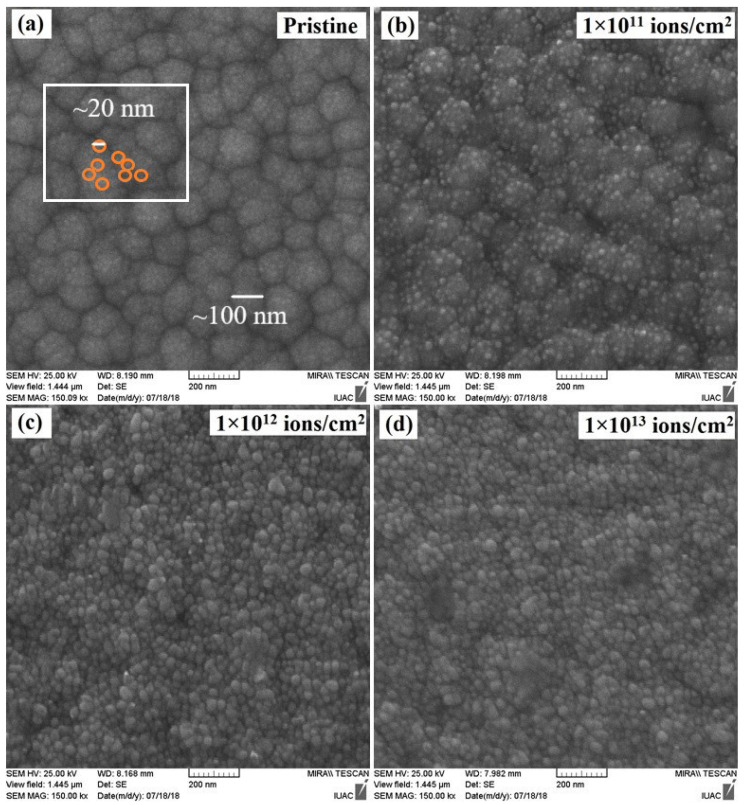
FESEM images of In_2_(Te_0.98_Se_0.02_)_3_ films (**a**) pristine, (**b**) 1E11, (**c**) 1E12, and (**d**) 1E13.

**Figure 4 nanomaterials-12-03782-f004:**
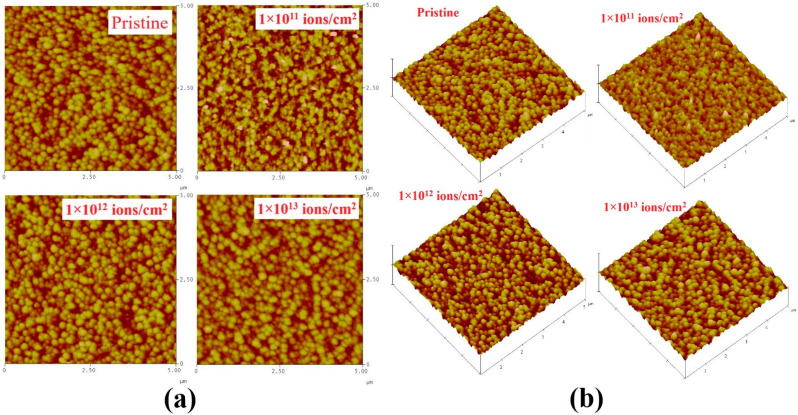
The 2D (**a**) and 3D (**b**) AFM images of In_2_(Te_0.98_Se_0.02_)_3_ thin films.

**Figure 5 nanomaterials-12-03782-f005:**
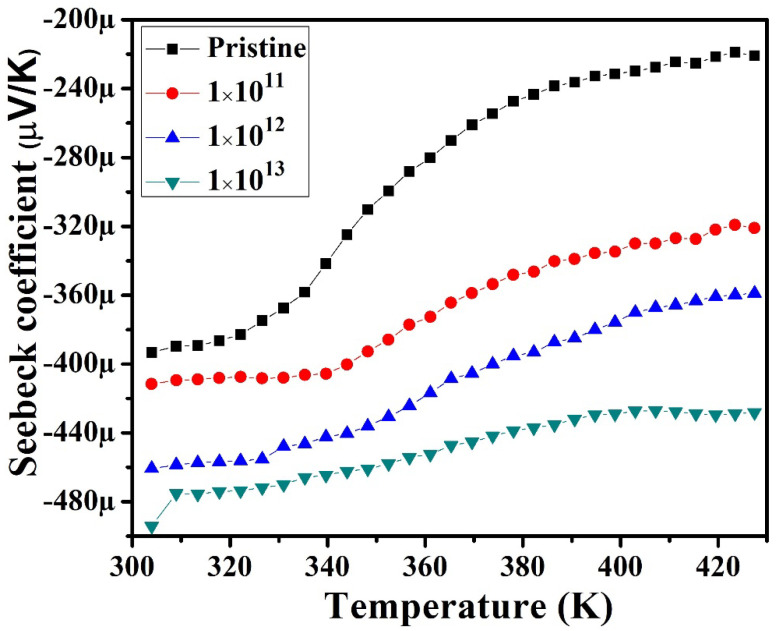
Temperature-dependent Seebeck coefficients of pristine and irradiated In_2_(Te_0.98_Se_0.02_)_3_ films.

**Figure 6 nanomaterials-12-03782-f006:**
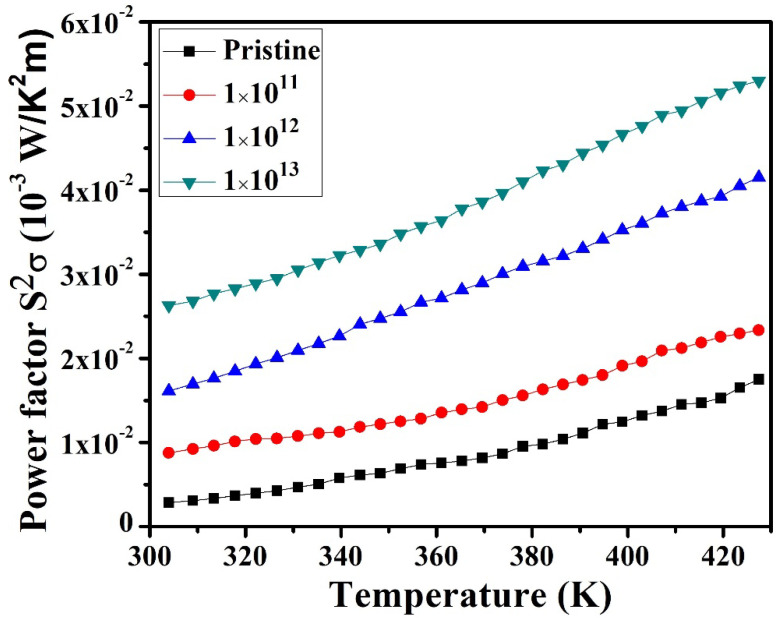
Temperature-dependent power factors (S^2^σ) of pristine and irradiated In_2_(Te_0.98_Se_0.02_)_3_ films.

**Figure 7 nanomaterials-12-03782-f007:**
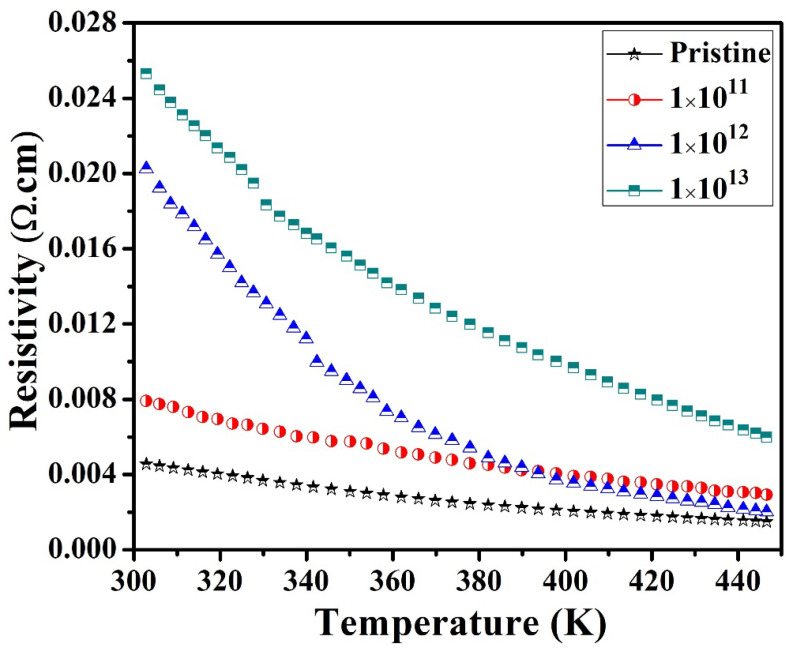
Electrical resistivities (*ρ*) of pristine and irradiated In_2_(Te_0.98_Se_0.02_)_3_ films.

**Figure 8 nanomaterials-12-03782-f008:**
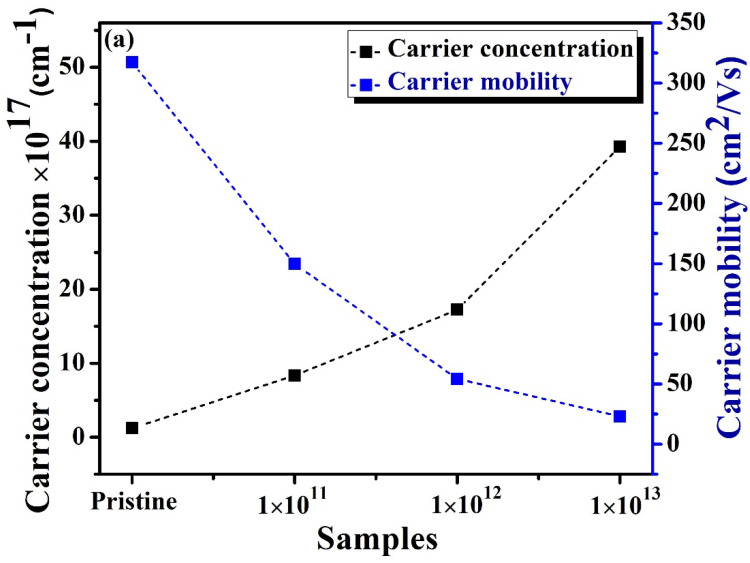
(**a**,**b**) Hall effect measurements of In_2_(Te_0.98_Se_0.02_)_3_ films.

**Table 1 nanomaterials-12-03782-t001:** Displays the calculated electronic energy loss (S_e_) using SRIM simulations.

In_2_(Te_0.98_Se_0.02_)_3_				
Ion	Ion Energy (MeV)	Se (eV/Å)		Range (µm)
		S_e_	S_n_	
Nickel	40	934	5.05	8.49
	60	1027	3.65	10.49
	80	1074	2.88	12.37
	100	1097	2.40	14.20
	120	1110	2.06	16.00
	150	1117	1.72	18.68

**Table 2 nanomaterials-12-03782-t002:** Elemental compositions of In_2_(Te_0.98_Se_0.02_)_3_ thin films.

In_2_(Te_1−x_Se_x_)_3_ (In: 40%, Te: 58%, Se: 2%)	Measured Elemental Composition (wt.%)					
	In	Te	Se	Si	C	O
Pristine	31.50	59.40	1.90	3.01	2.85	1.34
1E11 1E12	31.91	58.14	1.95	3.45	3.50	1.05
	31.63	58.80	1.87	3.63	3.25	0.82
1E13	31.89	58.92	1.79	3.54	3.17	0.69

**Table 3 nanomaterials-12-03782-t003:** Presents the Hall effect measurements of In_2_(Te_0.98_Se_0.02_)_3_ films.

In_2_(Te_0.98_Se_0.02_)_3_	Resistivity (Ω-cm)	Hall Mobility μH (cm^3^/V.s)	Carrier Concentration N_H_ (cm^−3^)	Hall Coefficient R_H_ (cm^3^/Coul.)
Pristine	0.073	317.09	1.21 × 10^17^	−40.18
1E11	0.125	149.78	8.35 × 10^17^	−35.54
1E12	0.155	54.20	17.24 × 10^17^	−21.13
1E13	0.198	23.12	39.27 × 10^17^	−16.23

## Data Availability

Data are available from the authors on request.

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
