# Peer review of "Effects of Heavy Ion Irradiation on the Thermoelectric Properties of In2(Te1−xSex)3 Thin Films"

_nanomaterials, 2022, doi:10.3390/nano12213782_

Round 1
Reviewer 1 Report
The manuscript presents a study of the effect of swift heavy ion irradiation on the thermoelectric properties of thin films of In-Te-Se, showing promising increase of the thermoelectric power factor. The results are clear, but I can recommend publication only after the following points have been considered:
-The study needs to be motivated much better: Why this particular material? Why such high energy irradiation? In particular, shoudn't lower energy ion beams i.e. simpler accelerators, do the job as well?
Introduction:
"the power factor quantifies the thermoelectric performance of a material". Misleading, as thermal conductivity matter as well (in ZT). Reword to say that it is an important part, (but not all). Also, make clear the definition of ZT in terms of the power factor.
-"In Se Te has unique properties.." big words. If you claim uniqueness you need to justify your claim: what is unique? Why is it promising for thermoelectrics in the first place?
-When talking about swift heavy ion irradiation" perhaps you can clarify which energies typically count as "swift", i.e. what is the usual definition of the terminology
"High energy deposition ..produces. exciting phenomena" in connection to the previous remark, how high does the energy need to be to produce the "exciting" phenomena? Would 1 MeV be enough? Also, exciting is not a good word (relates to emotions) change to "interesting" or some other milder word.
Materials and methods:
-Give the thickness of the films studied+substrate used.
-Any comment on stochiometry? How can you be sure that the stochiometry of the film is the same as the source? (Due to differing atomic vapor pressures)
-What kind of accelerator was used to produce the 120 MeV Ni beam?
-I would be very helpful if you could present SRIM simulations (or equivalent, its free software) to study how the ions penetrate the film-substrate and where the energy is mostly deposited?
Results and discussion
-Any explanation why you see the peaks of (105) and (120) planes and not others?
-You must define S_e , and explain how the value 1110 eV/å was calculated.
"Debye-Scherrer"
-"The dislocation density in 1E11 was lower" How do you know this at all?
"SHI deposition on target materials forms defects that are called thermal spikes by electron-electron coupling" This sentence is not understandable. Rephrase.
"Standard surface roughness (Ra)" perhaps this is standard with AFMs but it is not clear to me what it is (unlike the rms roughness)
-Is there correlation between the SEM and AFM images? Pristine samples especially look different to me
-The AFM images need a x-y scale bar
Measurement setup for S and \sigma needs to be explained: Did you pattern the films somehow lithographically? Cause if you didn't how could you determine resistivity (conductivity)?
Are you sure you want to plot resistivity instead of conductivity? Conductivity would go directy into the power factor.
-Explain that S and sigma are changed in opposite ways with irradiation (S increases, sigma decreases) but that S winds for the power factor.
-How did you determined carrier mobility? From Drude model?
-Do you have any spaculation why the carrier concentration changes so much with irradiation?
-"Charge scattering owing to grain boundaries considerably improves the thermoelectric power of the irradiated materials" Can you elaborate how? In other words why does seebeck coefficient increase (in absolute value)
-Please fill in the sections about author contributions and funding
Reviewer 2 Report
The authors present experimental results of ion irradiation n thermoelectric materials. What really happens as a function of the dose is mostly explained by SEM images. Especially the explanation on lines 108-122 can be a lot improved by exactly telling why the crystallites first grow witl low doses but then anyway become smaller. A phenomenological modelling would have been musch more informative. The results seem quite positive and may warrant publication. There are no error bars in any of the measurements and it is not clear what would have been the outcome of comparing different samples produced with the same parameters, are the results of this rather robust ion bombardment reproducible? On line 35 there is probably some error in grammar by claiming larger values of thermal conductivity?
